# Head-to-Head Comparison of High-Performance Liquid Chromatography versus Nuclear Magnetic Resonance for the Quantitative Analysis of Carbohydrates in Yiqi Fumai Lyophilized Injection

**DOI:** 10.3390/molecules28020765

**Published:** 2023-01-12

**Authors:** Yuesheng Xie, Dayong Zheng, Ting Yang, Zhenzhen Zhang, Wenwu Xu, Houru Liu, Wei Li

**Affiliations:** School of Pharmacy, North China University of Science and Technology, 21 Bohai Road, Caofeidian District, Tangshan 063210, China

**Keywords:** carbohydrates, qNMR, HPLC, quantitation, Yiqi Fumai lyophilized injection

## Abstract

Carbohydrate analysis can be used as a standard analysis for quality control of industries of plants, foods and pharmaceuticals. Quantitative ^1^H NMR spectroscopy (qNMR) is an excellent alternative to chromatography-based mixture analysis. However, the application of qNMR in sugar analysis has rarely been reported. In this study, the performance of qNMR in sugar analysis was investigated and compared with the results from HPLC analysis. A head-to-head comparison of qNMR (internal and external standard methods) versus HPLC (PMP pre-column derivatization HPLC, HPLC-RID and HPLC-ELSD) based on quantitative analysis of four carbohydrates (fructose, glucose, sucrose and maltose) in Yiqi Fumai lyophilized injection (YQFM) is presented. Both assays showed similar performance characteristics, including linearity range, accuracy, precision and recovery, and analysis times of less than 30 min/sample. After methodological validation, both qNMR and HPLC have good accuracy, precision and stability. Indeed, the qNMR method is simple, sensitive and rapid in quantifying the four sugars. By analysis of variance (ANOVA) for sugar content with HPLC and qNMR methods, we demonstrated that the two analytical methods had no significant difference and could be used interchangeably for the quantitative analysis of carbohydrates.

## 1. Introduction

Plants can assimilate atmospheric carbon dioxide into carbohydrates, and sugars as nutrients that humans can efficiently metabolize [1]. Sugars typically refer to a category of carbohydrates that includes monosaccharides like fructose and glucose, and disaccharides, such as sucrose and maltose [2]. The analyses of sugars are the most widespread chemical analyses performed in foods, beverages, forages, pharmaceuticals and other industries [3]. Meanwhile, carbohydrate analysis is standard practice in these industries and is often used for quality assurance [4].

Currently, several methods of sugar analysis exist and a new generation of instruments has emerged [5,6,7]. Ultraviolet (UV)-based detection is widely used in high-performance liquid chromatography (HPLC), mainly because of its ease of operation and relative sensitivity [8]. Due to the absence of UV chromophores in sugars, derivatization of the compounds may be used for sugar analysis. Besides, HPLC with a refractive index detector (HPLC-RID) is the most commonly used and conventional chromatographic technique for determining sugars in various matrices [9]. Moreover, HPLC coupled with an evaporative light-scattering detector (ELSD) is not limited by composition, the flow rate of the mobile phase and temperature, and may detect most compounds that are less volatile than the mobile phase. HPLC-ELSD is considered a good solution for analyzing carbohydrates in actual samples such as plants and foods [7]. These detection methods have some similarities, but the sensitivity and stability of specific analytes will vary when using different detectors. In addition, high-resolution nuclear magnetic resonance (NMR) spectroscopy has been increasingly applied to the quantitative analysis of analytes [10]. NMR analysis is based on the quantitative characterization of the ^1^H NMR signal intensities, which correspond to the number of protons in the molecule, and the use of an appropriate internal standard (IS) makes qNMR very attractive for purity determination [10,11]. NMR was applied to determine analytes in natural products, foods, beverages and pharmaceuticals because qNMR could be performed in a single, rapid and non-destructive measurement [10,12].

Although there are many precedents demonstrating the usefulness of NMR techniques [13], qNMR is not routinely used to quantify sugars. From a practical point of view, quantitative phytochemical analysis in quality assurance is always limited by resources, including sample material, access to instrumentation, and investigation time. Therefore, alternative technologies with low sample throughput, high personnel demand, and extensive sample workup by alternative approaches are core requirements for such procedures. Although both qNMR and HPLC are well-established and ubiquitous techniques, head-to-head comparisons for the quantitative analysis of carbohydrates have rarely been studied.

Traditional Chinese medicine (TCM) is used in pharmaceuticals and daily dietary supplements. Yiqi Fumai lyophilized injection (YQFM) derives from the recombination of the TCM formula “Sheng Mai SAN (SMS; an herbal powder)”, composed of red ginseng, Radix of *Ophiopogon japonicus* and Fructus of Schisandra chinensis, which has been applied clinically to efficaciously and safely treat chronic heart failure (CHF) [14,15,16]. These three herbs have a long history of edible and officinal in China. The main chemical constituents of these three medicinal substances are saponins, flavonoids, lignans and sugars [17,18,19]. The studies on saponins, flavonoids and lignans are relatively extensive, but the research on sugars is relatively scarce [18,20]. There is a lack of systematic research on the quantitative methods of sugar composition.

In this study, four sugar components (fructose, glucose, sucrose and maltose) in YQFM were quantified by PMP (1-Phenyl-3-methyl-5-pyrazolone) pre-column derivatization-HPLC (PMP-HPLC), HPLC-ELSD, HPLC-RID, qNMR external standard method (ESM) and qNMR internal standard method (ISM). The three modes of HPLC were compared with the two methods of qNMR. Our results showed that the five practices were not significantly different, and could be used for the quantitative analysis of the four sugars of YQFM. However, compared with HPLC, the determination of carbohydrates by NMR is more straightforward and faster, and it can accurately determine the sugars in herbs, providing a reference for the quality control of sugars and basic research of chemical substances, as well as laying a foundation for clinical drug evaluation and food safety. The present study provides a new approach and thought for determining sugars in the future.

## 2. Results

### 2.1. Validation of Quantitative HPLC Analysis

To allow head-to-head comparison of the qNMR analysis of carbohydrates (Figure 1) with more conventional analytical techniques, HPLC (PMP-HPLC-UV, HPLC-RID and HPLC-ELSD) assays were established. As shown in Figure 2, which are typical chromatograms, glucose and maltose were well separated by PMP-HPLC, and all four analytes were well resolved by HPLC-RID and HPLC-ELSD. For the PMP-HPLC-UV method, the linearity range of glucose and maltose was 0.01–1.00 mg/mL with correlation coefficients higher than 0.9998 (Table 1). Intra-day variations were less than 0.7%, and inter-day variations did not exceed 1.2%. The repeatability was less than 2.3%, and the stability was less than 1.9%. Sample recovery was less than 3.3% (Table 2). By HPLC-RID analysis, the linearity ranges for fructose, glucose, sucrose and maltose were 0.10–2.08, 0.10–2.13, 0.10–2.07 and 0.10–2.05 mg/mL, respectively (Table 1). Within these ranges, the calibration function correlation coefficients were better than 0.9993 for all analytes. Intra-day variations were less than 1.3%, and inter-day variations were no more than 2.2%. The repeatability was less than 2.2%, and the stability was less than 1.5%. Sample recovery was less than 3.2% (Table 2). Meanwhile, the linearity range of fructose was 0.10–2.00 mg/mL by the HPLC-ELSD method, while the linearity range of glucose, sucrose and maltose was 0.20–4.00 mg/mL (Table 1). Calibration correlation coefficients of all analytes were better than 0.9994. Intra-day variations were less than 1.7%, and inter-day variations did not exceed 2.6%. The results of repeatability, stability and recovery were listed in Table 2, which were all acceptable and reliable.

### 2.2. Validation of qNMR Analysis

^1^H-NMR proton signal peaks are mainly distributed in δ 0.5–9.0, and divided into three characteristic regions, including fatty acids, amino acids and organic acids in the high field region (δ 0.5–3.0); sugars in the middle field region (δ 3.0–5.5); and aromatic compounds in the low field region (δ 5.5–9.0). The qNMR spectra of fructose, glucose, sucrose and maltose are shown in Figure 3. The detailed attribution information of qNMR spectra of the four sugars is listed in Table 3.

qNMR analysis is achieved by comparing the intensities of specific absorption peaks. The peak with good separation and no overlap with other peaks should be selected as the quantitative peak [11]. Glucose has a well-resolved proton peak at *δ* 5.18 (d, 3.7 Hz), but this peak coincides with the proton peak of maltose. Besides, for the proton peak at *δ* 4.59 (d, 7.9 Hz), the left peak is disturbed, but the single peak on the right is unaffected, which can be quantified as characteristic quantitative hydrogen. Therefore, the proton peak at *δ* 4.59 (d, 7.9 Hz) was chosen as the distinct quantitative peak of glucose (Figure 3).

The analytical precision, repeatability, stability and recovery were validated by qNMR ISM and ESM. As listed in Table 4, the intra-day variation of the ISM was less than 2.2%, and the intra-day variation did not exceed 2.9%. The repeatability was less than 3.6% and the stability was less than 2.6%. Sample recovery was less than 2.5%. Similarly, the intra-day variation of the ESM was less than 2.2%, and the intra-day variation did not exceed 3.1%. The repeatability was less than 3.4% and the stability was less than 2.5%. Sample recovery was less than 2.5%. According to the standard curve of ESM, the linearity range of fructose was 1.36–21.72 mg/mL, the linearity range of glucose and sucrose was 0.33–5.22 mg/mL, and the linearity range of maltose was 0.22–3.54 mg/mL. The calibration function correlation coefficients were better than 0.9996 for all analytes (Table 5).

### 2.3. Sample Analysis

Validated HPLC and qNMR methods were then applied to the quantification of ten batches of samples. As listed in Table 6, the content of compounds varied little and the proportion was relatively constant, with higher content of fructose (~12.98%) and lower maltose content (~2.65%) in YQFM. Additionally, one-way ANOVA was applied to evaluate HPLC and NMR methods, and the sugar content in 10 batches of YQFM samples was used as the evaluation index. Table 7 summarizes the comparison of various results for the difference in sugar contents by different methods. The selected variables were not significantly other among groups, indicating that the null hypothesis might be accepted. The results showed that the two techniques were equivalent in quantitatively evaluating fructose, glucose, sucrose and maltose. No significant differences (F < F critical) were found in the determination of sugar contents (Table 7). This study also demonstrates that the established qNMR method could be used for the accurate quantification of sugars.

## 3. Discussion

Because the sugar component has no UV absorption, it needs to be derivatized and then analyzed by HPLC. PMP, a commonly used derivatization reagent, could undergo a condensation reaction with reducing sugar under alkaline conditions, and the reaction product has a strong absorption at 245 nm [21]. Besides, fructose and other non-reducing sugars, such as sucrose, can not react with PMP under studied conditions due to steric hindrance or lack of aldehyde group on their molecules [22], and there are no corresponding chromatographic peaks in the chromatogram after derivatization. While PMP-HPLC can quantify carbohydrates in an accessible manner under mild reaction conditions, the determination of sugars has limitations. In addition, we evaluated the effect of different concentrations of phosphate buffer (0.01%, 0.02%, 0.05% and 0.10%) on the separation. With phosphate buffer at the concentration of 0.05% and 0.10%, the maltose peak appeared tailing, and the higher phosphate concentration led to the increase of column pressure. At the concentration of 0.01%, the glucose peaks could not be completely separated, while at 0.02%, the peak types of glucose and maltose were better. Moreover, the different proportions of acetonitrile-0.02% phosphate solution (25:75, 20:80, 17:83 and 15:85, *v*/*v*) were evaluated, and we found that the chromatographic peak resolution of each component was better at the ratio of 17:83.

Mobile phases of acetonitrile-water or methanol–water systems were investigated for HPLC-RID and HPLC-ELSD. The acetonitrile-water system showed a rapid peak time, reasonable resolution of measured peaks and sharp peak shape. Meanwhile, different ratios of acetonitrile-water (80:20, 75:25, 65:35, *v*/*v*) were also explored, and finally selected the ratio of acetonitrile-water (75:25, *v*/*v*) as the mobile phase. Furthermore, the chromatographic columns were evaluated for HPLC-RID and HPLC-ELSD analysis, including the Agilent XDB-C_18_ column, Shodex Asahipak NH_2_P-50 column and unitary NH_2_ column. The unitary NH_2_ column could not effectively separate the chromatographic peaks of glucose and maltose, and the Agilent XDB-C_18_ column could not analyze the sugars. In contrast, the Shodex Asahipak NH_2_P-50 column was better for separating glucose and maltose. Therefore, Shodex Asahipak NH_2_P-50 chromatographic column was chosen as the chromatographic column for separation.

In a series of preliminary qNMR experiments, several deuterium-labeled solvents, including CDCl_3_, D_2_O, CD_3_OD, C_5_D_5_N and their mixtures, were tested as sample solvents. The obtained spectrum indicated that D_2_O: CD_3_OD (1:1, *v*/*v*) provided sufficient solubility for the compounds. D_2_O: CD_3_OD (1:1, *v*/*v*) was chosen as the solvent. The critical factors for qNMR are good separation of resonances and clear baselines in the relevant spectral regions. TSP has good solubility and stability in CD_3_OD, and does not overlap or interact with the proton peaks of analytes, with stable properties, a single signal peak, and the chemical offset close to zero [23]. Thus, TSP was chosen as the IS for qNMR.

The comparison of the data from HPLC and qNMR proves that the two techniques are equivalent in quantifying the content of four sugars in YQFM. However, qNMR is faster and less destructive to the sample than conventional HPLC. From an economic point of view, the two methods are also comparable when used for routine analysis. Although NMR equipment is generally more expensive than chromatography instrumentation, qNMR has a lower cost per sample due to reduced solvent usage and consumables requirements [24]. Moreover, the sample detection time is also comparable, and dozens of samples could be run every day [24].

## 4. Materials and Methods

### 4.1. Chemicals

Fructose (purity ≥ 99.8%), D-glucose anhydrous (purity ≥ 99.9%), sucrose (purity ≥ 99.8%) and maltose (purity ≥ 99.8%) were purchased from National Institutes for Food and Drug Control (Beijing, China), and their chemical structures were shown in Figure 1. 1-Phenyl-3-methyl-5-pyrazolone (PMP, analytical grade and purity ≥ 99.5%) was obtained from Tianjin kemio Chemical Reagent Co., Ltd. (Tianjin, China). Absolute ethanol (analytical grade) was purchased from Tianjin Zhiyuan Chemical Reagent Co., Ltd. (Tianjin, China). HPLC-grade acetonitrile and methanol were purchased from Merck (Darmstadt, Germany). Deuterium oxide (D_2_O, 99.9 atom % D), methanol-d_4_ (CD_3_OD, 99.8 atom % D), and 3-(Trimethylsilyl)propionic-2,2,3,3-d_4_ acid sodium salt (TSP-d_4_, 98 atom % D) were purchased from Sigma-Aldrich Trading Co., Ltd. (Shanghai, China). YQFM samples of 10 batches (20190409, 20190514, 20190707, 20190906, 20191101, 20201005, 20200809, 20201105, 20210810 and 20210909) were obtained from Tianjin Tasly Pride Pharmaceutical Co., Ltd. (Tianjin, China), numbered S1~S10.

### 4.2. Preparation of Standard Solutions and Samples for the HPLC Method

*PMP-HPLC*: Glucose and maltose were dissolved in 50% methanol–water solution to prepare mixed standard solutions with a concentration of 1.00 mg/mL, respectively.

*HPLC-RID and HPLC-ELSD*: Fructose, glucose, sucrose and maltose were dissolved in a 50% methanol–water solution to prepare mixed standard solutions with concentrations of 2.08, 2.13, 2.08 and 2.05 mg/mL for HPLC-RID analysis, respectively. For HPLC-ELSD analysis, the mixed standard solutions of fructose, glucose, sucrose and maltose were prepared at concentrations of 0.30, 0.60, 0.60 and 0.60 mg/mL, respectively.

*Sample preparation*: The sample solutions of the three methods were prepared by precisely weighing 40 mg of YQFM samples, dissolving them in a 50% methanol–water solution, shaking well and ultrasonic to obtain the 20 mg/mL sample solution. All samples were stored at 4 °C during the study.

### 4.3. Sample Preparation for qNMR Method

TSP was dissolved in D_2_O and CD_3_OD solution (1:1, *v*/*v*) to prepare an internal standard (IS) at a concentration of 0.2 mg/mL. YQFM samples (40 mg) were dissolved in an IS solution to obtain a sample solution of 80 mg/mL. Fructose, glucose, sucrose and maltose were dissolved in an IS solution to prepare mixed standard solutions with concentrations of 21.72, 5.22, 5.22 and 3.54 mg/mL, respectively. All samples were stored at 4 °C during analysis.

### 4.4. HPLC Conditions

#### 4.4.1. Derivatization and PMP-HPLC Conditions

Add 50 μL NaOH (0.3 mol/L) and 50 μL PMP-methanol solution (0.5 mol/L) to 50 μL standard or sample solution, and incubate at 70 °C for 30 min. The reaction solution was neutralized with 50 μL HCl (0.3 mol/L), and then extracted with 200 μL of chloroform for three times. The final aqueous phase was collected and filtered using a 0.45 μm filter for PMP-HPLC analysis.

PMP-HPLC analysis was performed using the LS Solution system (Shimadzu, Japan) with a quaternary pump, autosampler, column thermostat, and ultraviolet detector. Agilent XDB-C_18_ column (250 mm × 4.6 mm, 5 μm) was used as the stationary phase. Acetonitrile-0.02% phosphate solution (17:83, *v*/*v*) was used as the mobile phase. The detection wavelength is 245 nm.

#### 4.4.2. HPLC-RID and HPLC-ELSD Conditions

HPLC-RID analysis was performed using the CAG Bootp Serve system (Agilent, Santa Clara, CA, USA) with a quaternary pump, autosampler, column thermostat and differential detector. HPLC-ELSD assays were implemented using the Empower3 system (Waters, Milford, MA, USA) with a quaternary pump, autosampler, column thermostat and Waters ELSD 2424 evaporative light-scattering detector. The drift tube temperature was 40 °C and the gain was 10. Shodex Asahipak NH_2_P-50 column (250 mm × 4.6 mm, 5 μm) was used as the separation column, and acetonitrile-water (75:25, *v*/*v*) was used as the mobile phase.

The methods of PMP-HPLC, HPLC-RID and HPLC-ELSD were used as an isocratic elution. The column temperature was 30 °C, the flow rate was 1.0 mL/min, and the injection volume was 10 μL.

### 4.5. qNMR Experimental Parameters

qNMR spectra were acquired on the Bruker Avance II 600 spectrometer (Bruker, Switzerland) equipped with a 5 mm prodigy cryoprobe. qNMR spectrum was obtained using a Bruker pulse program “zgcppr” under the following settings: pulse angle = 90°; probe temperature = 298 K; relaxation delay (D1) = 8 s; acquisition time = 2.99 s; FID (free induction decay) data point = 64 K; spectral width = 12 ppm; number of scans = 64. For pre-saturation experiments, the transmitter offset was manually set at *δ* 4.7 ppm to achieve the best suppression of the residual water signal. The obtained FIDs were all Fourier transformed to produce spectra with 128 K data points (zero padding). Manual phase correction and automatic polynomial baseline correction are typically used. The chemical shift value refers to the calibration standard (TSP) signal.

### 4.6. Calibration and Validation

HPLC and qNMR methods were validated in line with the FDA Guidelines (draft 2013 version). The standard stock solution was prepared by dissolving the compounds fructose, glucose, sucrose and maltose in D_2_O: CD_3_OD (1:1, *v*/*v*) containing IS (TSP, 0.2 mg/mL). Stock solution I (fructose: 21.72 mg/mL; glucose: 5.22 mg/mL; sucrose: 5.22 mg/mL; maltose: 3.54 mg/mL, respectively) was obtained, and diluted the solution according to the volume ratio of stock solution I to IS solution (1:2, 1:4, 1:8 and 1:16, *v*/*v*). The calibration curve was generated by plotting the peak areas of the analytes versus concentrations of fructose, glucose, sucrose and maltose in HPLC, as well as the peak area ratio and concentration ratios of fructose, glucose, sucrose and maltose in qNMR. The correlation coefficient (*r*) was calculated by linear regression analysis. Intra- and inter-day accuracy was assessed by parallel injections of the mixed standard solutions over six consecutive days. The stability was measured at 0, 2, 4, 8, 10 and 12 h, respectively. The lower limit of quantitation (LLOQ) is the lowest analytical concentration with signal-to-noise (S/N) ≥10. Six samples in parallel were analyzed to determine repeatability. Sample spiking recovery was the addition of a known amount of analyte to the sample before sample preparation, and all samples were performed in six replicates.

### 4.7. Statistics

Data analysis for analyte concentration verification and calculation was completed in the Microsoft Excel 2011 (Raymond, Washington, DC, USA) spreadsheet. Data were analyzed using the D’Agostino and Pearson omnibus normality test to determine whether assumptions of one-way analysis of variance (ANOVA) were met. If assumptions were met, ANOVA was performed. Statistical analysis was performed using SPSS statistics (SPSS 22.0 software).

## 5. Conclusions

Both assays showed similar performance characteristics, including linearity range, accuracy, precision and recovery, and analysis times of less than 30 min/sample. After methodological validation, both qNMR and HPLC have good accuracy, precision and stability. Comparing HPLC and NMR, both methods could accurately quantify fructose, glucose, sucrose and maltose in YQFM. There was no significant difference in the results of sugar contents measured by the two techniques. HPLC and qNMR may be used interchangeably and provide an essential reference for the determination of sugars in plants, foods or pharmaceutical preparations.

## Figures and Tables

**Figure 1 molecules-28-00765-f001:**
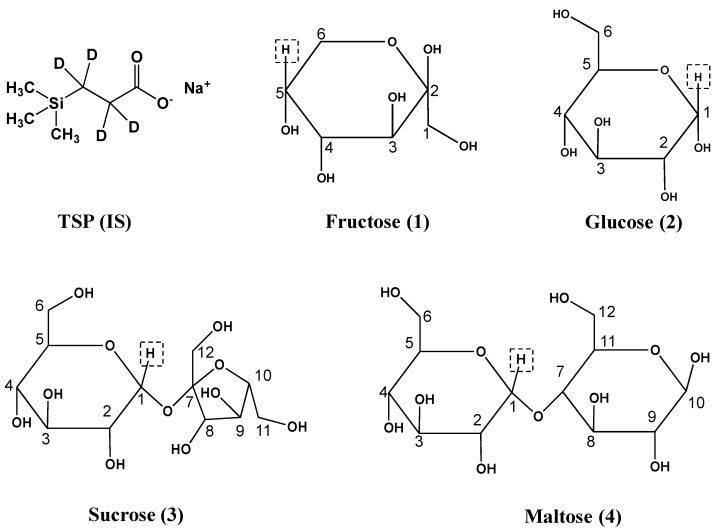
Structures of TSP (internal standard, IS) and sugars. The analyte fructose, glucose, sucrose and maltose were numbered 1–4, respectively. qNMR resonances of highlighted protons were used for qNMR-based analyte quantitation.

**Figure 2 molecules-28-00765-f002:**
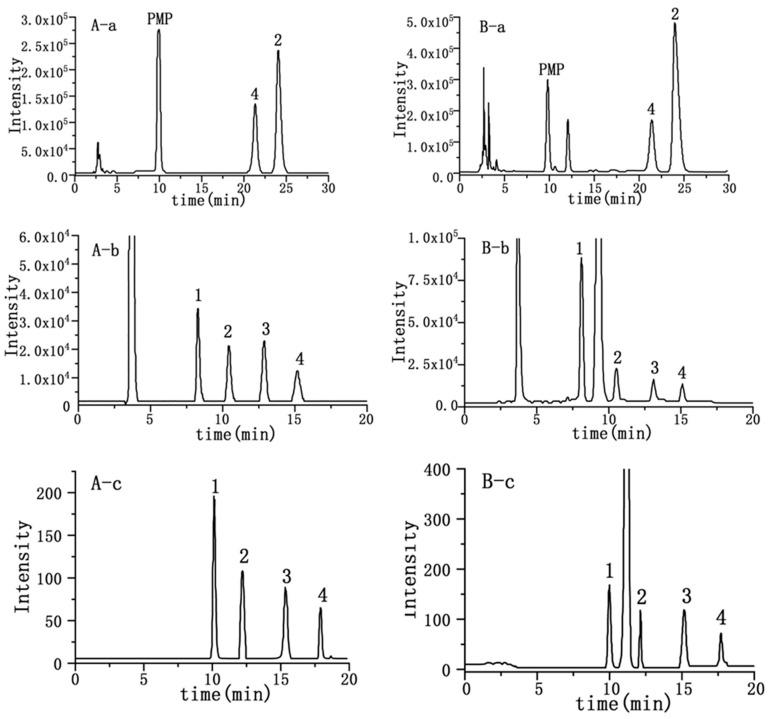
Chromatograms of standard solution (**A**) and sample solutions (**B**) of sugars from YQFM for PMP-HPLC (**a**), HPLC-RID (**b**) and HPLC-ELSD (**c**). Numbers 1–4 correspond to fructose, glucose, sucrose and maltose, respectively.

**Figure 3 molecules-28-00765-f003:**
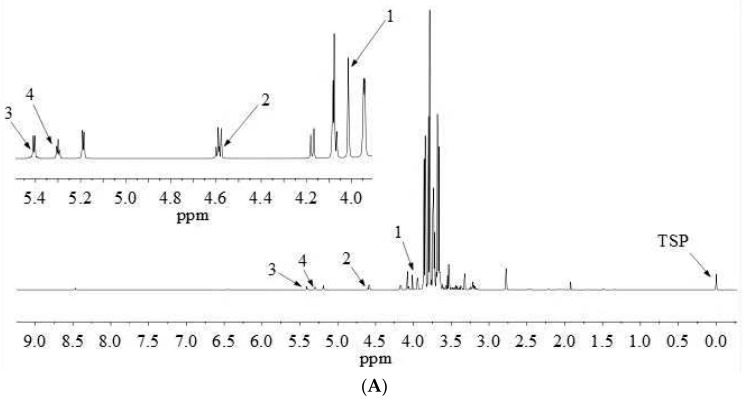
qNMR spectrum of YQFM at the spectral range from δ = 0.0 to 9.0 ppm with an enlarged view on the spectral range from δ = 4.0 to 5.5 ppm, measured with a 600 MHz spectrometer (**A**). Highlighted inserts show NMR peaks (top in sample) used for analyte quantitation, Numbers 1–4 correspond to fructose, glucose, sucrose and maltose, respectively (**B**).

**Table 1 molecules-28-00765-t001:** HPLC-based calibration function parameters for compounds, including regression equations, correlation coefficients (*r*) and linearity range.

Method	Compound	*Y* (Peak Area) = kc_A_ + d	*r*	Linearity Range (mg/mL)	LLOQ (mg/mL)
PMP-HPLC	Glucose	*Y* = 17,739*X* + 6829.8	0.9998	0.01~1.00	0.01
	Maltose	*Y* = 9116.4*X* + 1027.0	1.0000	0.01~1.00	0.01
HPLC-RID	Fructose	*Y* = 103,246*X* + 6774.8	0.9998	0.10~2.08	0.10
	Glucose	*Y* = 116,724*X* + 2647.4	0.9993	0.10~2.13	0.10
	Sucrose	*Y* = 83,425.0*X* + 8358.1	0.9994	0.10~2.07	0.10
	Maltose	*Y* = 67,678.0*X*−2551.7	0.9997	0.10~2.05	0.10
HPLC-ELSD	Fructose	*Y* = 1.3246X +0.6774	0.9999	0.10~2.00	0.10
	Glucose	*Y* = 1.3425X + 0.8358	0.9994	0.20~4.00	0.20
	Sucrose	*Y* = 1.0548X−0.6375	0.9995	0.20~4.00	0.20
	Maltose	*Y* = 1.1267X−0.2551	0.9996	0.20~4.00	0.20

**Table 2 molecules-28-00765-t002:** HPLC assay accuracy for compounds. Inter-day recovery values (n = 6) expressed in percentage amount added (mean ± RSD).

Method	Compound	Recovery (%±RSD)
PMP-HPLC	glucose	99.4 ± 3.3
	maltose	102.4 ± 2.9
HPLC-RID	fructose	100.1 ± 0.4
	glucose	101.4 ± 3.2
	sucrose	97.3 ± 2.5
	maltose	99.7 ± 2.8
HPLC-ELSD	fructose	98.6 ± 1.2
	glucose	100.6 ± 1.5
	sucrose	99.6 ± 3.0
	maltose	100.1 ± 2.5

**Table 3 molecules-28-00765-t003:** qNMR signal information (600 MHz, with MeOH-d_4_: D_2_O (50:50, *v*/*v*) as solvent for compounds) for NMR resonances used within the ^1^H-NMR experiments for sugars from YQFM quantitation.

Compound	δ_H_ (ppm)	Multiplicity, J_HH_ (Hz)
fructose	4.16	d, *J* = 7.8 Hz
	**4.01**	d, *J* = 3.1 Hz
	3.98	d, *J* = 1.8 Hz
	3.81	d, *J* = 0.8 Hz
	3.77	d, *J* = 3.1 Hz
	3.68	d, *J* = 2.0 Hz
glucose	5.18	d, *J* = 3.7 Hz
	**4.59**	d, *J* = 7.9 Hz
	3.84	d, *J* = 2.4 Hz
	3.46	t, *J* = 2.1 Hz
	3.40	d, *J* = 2.2 Hz
	3.20	d, *J* = 1.4 Hz
sucrose	**5.41**	d, *J* = 3.7 Hz
	4.04	t, *J* = 8.3 Hz
	3.76	d, *J* = 2.6 Hz
	3.68	t, *J* = 2.0 Hz
	3.44	d, *J* = 1.0 Hz
maltose	**5.29**	d, *J* = 3.7 Hz
	5.20	d, *J* = 3.7 Hz
	4.60	d, *J* = 7.8 Hz
	3.90	t, *J* = 2.4 Hz
	3.86	d, *J* = 1.4 Hz
	3.25	d, *J* = 1.6 Hz

Numbers in bold represent the selected quantitative peak. d, double peak; t, triple peak.

**Table 4 molecules-28-00765-t004:** qNMR assay accuracy for compounds. Inter-day recovery values (n = 6) expressed in percentage amount added (mean ± RSD).

Method	Compound	Recovery (% ± RSD)
qNMR ISM	fructose	96.5 ± 2.5
	glucose	101.2 ± 2.2
	sucrose	97.1 ± 2.3
	maltose	100.8 ± 2.3
qNMR ESM	fructose	97.5 ± 2.6
	glucose	102.2 ± 1.2
	sucrose	100.3 ± 3.3
	maltose	97.6 ± 2.1

**Table 5 molecules-28-00765-t005:** qNMR ESM-based calibration function parameters for compounds, including regression equations, correlation coefficients (*r*) and linearity range.

Compound	*Y* (Peak Area) = kc_A_ + d	*r*	Linearity Range (mg/mL)	LLOQ (mg/mL)
Fructose	*Y* = 0.2013*X* + 0.0200	0.9999	1.36~21.72	1.36
Glucose	*Y* = 0.4489*X* − 0.0850	0.9999	0.33~5.22	0.33
Sucrose	*Y* = 0.2679*X* − 0.0237	0.9996	0.33~5.22	0.33
Maltose	*Y* = 0.1116*X* − 0.0007	0.9998	0.22~3.54	0.22

**Table 6 molecules-28-00765-t006:** Results of HPLC and qNMR determination of sugar content (%) in YQFM samples (mean ± RSD, n = 3).

Compound	Samples	Method				
		PMP -HPLC	HPLC -RID	HPLC -ELSD	qNMR ISM	qNMR ESM
Fructose	S1	/	14.382 ± 0.043	14.398 ± 0.047	14.375 ± 0.042	13.928 ± 0.035
	S2	/	12.795 ± 0.030	12.776 ± 0.024	12.716 ± 0.030	12.609 ± 0.028
	S3	/	12.675 ± 0.047	12.686 ± 0.039	12.417 ± 0.032	12.528 ± 0.026
	S4	/	11.862 ± 0.036	11.874 ± 0.021	12.469 ± 0.024	12.849 ± 0.035
	S5	/	13.627 ± 0.034	13.608 ± 0.036	13.452 ± 0.028	14.013 ± 0.031
	S6	/	11.893 ± 0.020	11.876 ± 0.033	12.675 ± 0.016	12.817 ± 0.011
	S7	/	13.265 ± 0.031	13.273 ± 0.043	11.513 ± 0.016	12.073 ± 0.024
	S8	/	12.832 ± 0.037	12.840 ± 0.021	13.529 ± 0.030	13.316 ± 0.022
	S9	/	13.041 ± 0.032	13.027 ± 0.035	12.891 ± 0.027	13.012 ± 0.030
	S10	/	12.647 ± 0.026	12.641 ± 0.034	14.107 ± 0.026	13.816 ± 0.031
Glucose	S1	3.075 ± 0.013	3.082 ± 0.017	3.076 ± 0.013	3.045 ± 0.016	3.217 ± 0.018
	S2	3.071 ± 0.016	3.068 ± 0.012	3.053 ± 0.011	3.472 ± 0.012	3.731 ± 0.009
	S3	3.032 ± 0.014	3.016 ± 0.008	3.026 ± 0.010	3.159 ± 0.010	3.263 ± 0.006
	S4	3.419 ± 0.007	3.431 ± 0.018	3.427 ± 0.015	3.721 ± 0.011	3.857 ± 0.015
	S5	3.637 ± 0.011	3.654 ± 0.014	3.647 ± 0.016	3.186 ± 0.014	3.504 ± 0.008
	S6	3.032 ± 0.010	3.025 ± 0.006	3.036 ± 0.013	3.597 ± 0.006	3.062 ± 0.011
	S7	3.416 ± 0.012	3.426 ± 0.012	3.432 ± 0.009	3.258 ± 0.010	3.146 ± 0.007
	S8	3.357 ± 0.020	3.373 ± 0.007	3.357 ± 0.011	3.489 ± 0.013	3.322 ± 0.009
	S9	3.673 ± 0.013	3.635 ± 0.012	3.626 ± 0.005	3.084 ± 0.010	3.140 ± 0.012
	S10	3.326 ± 0.010	3.347 ± 0.015	3.338 ± 0.014	3.721 ± 0.011	3.508 ± 0.007
Sucrose	S1	/	3.626 ± 0.010	3.638 ± 0.012	3.075 ± 0.014	3.184 ± 0.009
	S2	/	3.413 ± 0.016	3.442 ± 0.008	3.128 ± 0.015	3.319 ± 0.011
	S3	/	3.241 ± 0.007	3.235 ± 0.010	3.025 ± 0.005	3.137 ± 0.008
	S4	/	3.655 ± 0.017	3.646 ± 0.013	3.408 ± 0.009	3.627 ± 0.015
	S5	/	2.967 ± 0.011	2.982 ± 0.010	3.029 ± 0.005	3.176 ± 0.009
	S6	/	3.180 ± 0.016	3.174 ± 0.012	3.073 ± 0.010	3.112 ± 0.018
	S7	/	2.982 ± 0.009	2.967 ± 0.013	3.128 ± 0.016	3.035 ± 0.012
	S8	/	3.553 ± 0.021	3.564 ± 0.025	3.029 ± 0.011	3.247 ± 0.013
	S9	/	3.425 ± 0.013	3.441 ± 0.012	3.317 ± 0.011	3.245 ± 0.010
	S10	/	3.105 ± 0.007	3.123 ± 0.012	3.268 ± 0.010	3.409 ± 0.013
Maltose	S1	2.503 ± 0.008	2.525 ± 0.011	2.540 ± 0.009	2.582 ± 0.006	2.692 ± 0.010
	S2	2.926 ± 0.015	2.913 ± 0.019	2.907 ± 0.007	2.846 ± 0.012	2.918 ± 0.016
	S3	3.341 ± 0.022	3.338 ± 0.024	3.341 ± 0.028	3.022 ± 0.015	3.109 ± 0.016
	S4	2.782 ± 0.015	2.797 ± 0.020	2.806 ± 0.021	2.795 ± 0.015	2.817 ± 0.011
	S5	2.657 ± 0.018	2.638 ± 0.014	2.646 ± 0.011	2.687 ± 0.024	2.538 ± 0.014
	S6	2.281 ± 0.013	2.273 ± 0.011	2.257 ± 0.015	2.294 ± 0.011	2.226 ± 0.009
	S7	2.743 ± 0.017	2.758 ± 0.020	2.763 ± 0.018	2.746 ± 0.016	2.808 ± 0.021
	S8	2.586 ± 0.016	2.571 ± 0.021	2.546 ± 0.013	2.612 ± 0.018	2.589 ± 0.012
	S9	2.069 ± 0.011	2.082 ± 0.009	2.097 ± 0.007	2.024 ± 0.008	2.135 ± 0.005
	S10	2.265 ± 0.019	2.248 ± 0.016	2.263 ± 0.013	2.258 ± 0.015	2.209 ± 0.009

**Table 7 molecules-28-00765-t007:** Analysis of variance (ANOVA) of sugar content (Percent) in HPLC and qNMR methods ^a^.

Variable	*F* (4,45)	*F* (3,36)	*F* Critical (*p* = 0.05)	*p*
Fructose	/	0.157	1.95	0.924
Glucose	0.237	/	1.95	0.916
Sucrose	/	1.437	1.95	0.248
Maltose	0.013	/	1.95	0.999

^a^ The *F* values, with degrees of freedom (in parentheses), are the test values for each variable that are compared with the critical values (from tables) for the chosen probability (5%). *p* is the probability of the null hypothesis to be true.

## Data Availability

The data that support the findings of this study are available from the corresponding author upon reasonable request.

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
