# Peer review of "Head-to-Head Comparison of High-Performance Liquid Chromatography versus Nuclear Magnetic Resonance for the Quantitative Analysis of Carbohydrates in Yiqi Fumai Lyophilized Injection"

_molecules, 2023, doi:10.3390/molecules28020765_

Round 1

Reviewer 1 Report

The authors made a comparison between results collected on a HPLC and qNMR. Method based on a use of qNMR showed significantly better analytical characteristics, e.g. wider linear response range, but limit of quantification had not been extended to lower values, in fact, it had been increased at least twice and for glucose almost 200 times. On the other hand, qNMR methods have wider linear response range, especially for glucose, over 20 times, what is positive. Although, the authors made both HPLC and NMR analysis, gave results in Table 6 as well as statistical analysis in Table 7, they did not comment the results what could caused slight difference among HPLC and NMR results. I expect form authors giving an explanation for. It is very important to have such explanation, especially if an analyst intends to use the proposed NMR methods for carbohydrate determination. There are differences in used font, e.g. lines 30-31, 66-70 etc., please correct them.  The used literature can be "younger", there are lots of articles older than 10 years. After the authors do the mentioned changes, I would like to consider again the manuscript.    

Reviewer 2 Report

The manuscript "Head-to-head comparison of high-performance liquid chromatography versus nuclear magnetic resonance for the quantitative analysis of carbohydrates in Yiqi Fumai lyophilized injection” provides a straightforward comparison of qNMR and HPLC performance for quantitative analysis in real samples of pharmaceuticals. The study was well designed and described. In my view the statistical analysis of the results could be improved before publication in the following way:

1) Table 1, Table 5,  – please add uncertainties of slopes and intercepts, test whether intercept =0, use appropriate number of significant digits depending on the uncertainties.

2) page 7 l.141-152 – this paragraph is a repetition of data from tables, provide more concise summary of the data instead.

3) Table 7, This seems not entirely correct, depending on the compounds, there was either 4 or 5 methods therefore single number F(4,45) can’t be used. Also the data allows multiple-way ANOVA analysis with residuals calculated from n=3 repetitions of each results for each sample, possible interactions can be evaluated as well.

4) In the conclusions the costs of deuterated solvents vs the consumption of HPLC solvents can be compared to justify the use of each method. Potential limitations with regard to other matrices can be briefly discussed.

5) page 12 – 4.7. The information regarding whether the ANOVA assumptions were met should be provided.

Round 2

Reviewer 1 Report

The authors obeyed reviewer's comments and gave a satisfied explanations. Therefore, I would like to suggest acceptance of the manuscript.